# COVID-19 Vaccine Hesitancy in China: An Analysis of Reasons through Mixed Methods

**DOI:** 10.3390/vaccines11030712

**Published:** 2023-03-22

**Authors:** Yao Sun, Xi Li, Difan Guo

**Affiliations:** 1School of Journalism and Communication, Nanjing Normal University, Nanjing 210097, China; 2School of Journalism and Communication, Beijing Normal University, Beijing 100091, China

**Keywords:** vaccine hesitancy, willingness, COVID-19 vaccine, China, mixed methods

## Abstract

This study aims to investigate the causes of COVID-19 vaccine hesitancy among the Chinese population. The LDA model and content analysis were used to analyze the content of COVID-19 vaccine hesitancy expressed by the Chinese on Weibo from 2020 to 2022, the leading causes of vaccine hesitancy, and the changes in the reasons for vaccine hesitancy over time. The study found that when the Chinese expressed vaccine hesitancy, it usually involved themes such as information access (18.59%), vaccination services (13.91%), and physical illness (13.24%), and topics such as vaccination process (6.83%), allergic diseases (6.59%), and international news (6.43%). Constraints (35.48%), confidence (17.94%), and calculation (15.99%) are the leading causes of vaccine hesitancy on Weibo. These findings provide a comprehensive picture of how the Chinese express vaccine hesitancy in social media and the reasons and changes for vaccine hesitancy, which can help inspire public health experts, health organizations, or governments in various countries to improve the phenomenon of vaccine hesitancy.

## 1. Introduction

Due to the continued global spread of COVID-19 and the lack of a specific drug, vaccination has been the primary means of preventing and controlling COVID-19 in many countries [1]. The Chinese government provides the population with free COVID-19 vaccination services [2] and actively promotes “universal vaccination.” As of 31 December 2022, China has reported more than 3.4 billion cumulative doses of the COVID-19 vaccine [3]. China’s COVID-19 vaccination efforts are progressing rapidly, but some things could be improved in the implementation process. From the vaccine promoter’s perspective, COVID-19 vaccine communication and promotion efforts are vulnerable to hindrance from misinformation [4] and anti-vaccine rhetoric [5]. From the standpoint of vaccinators, although the government vigorously promotes positive features such as the “safety and efficacy of the vaccine”, the vaccination choice still rests with the public. In this process, some people are prone to vaccine hesitancy due to concerns about their underlying diseases [6], doubts about the reliability of vaccines, and other reasons [7]. Since China has a large population and complex national conditions, it is crucial to clarify the causes of vaccine hesitancy to implement vaccination successfully [8].

Vaccine hesitancy is the delay in receiving or refusing to vaccinate despite the availability of vaccine services. It was listed by the World Health Organization in 2019 as one of the top 10 global health threats [9]. Vaccine hesitancy leads to a negative willingness to vaccinate and is a significant factor in preventing infectious diseases and reducing vaccination rates [10]. Weibo is one of the China’s most popular social media platforms, with over 500 million active users [11,12,13]. User analysis report by the Weibo Company indicated bachelor’s degree or above users accounted for 76%, and the group aged 17 to 33 accounted for 83% of the total mobile users [14]. Since the COVID-19 outbreak, many Chinese have used social media to exchange information about the outbreak and vaccine, making Weibo one of the main venues for Chinese internet users to obtain and discuss health information [15,16,17]. Some studies confirm that social media does provide space for the public to express vaccine hesitations, which makes Weibo a mixed space containing public vaccine views and emotions [18]. Many active Weibo users and the massive amount of vaccine discussions also provide a sample source for our study of the causes of COVID-19 vaccine hesitancy in the Chinese population [19,20].

The 3C model (confidence, complacency, and constraints) has been commonly used in studies to analyze the psychological mechanisms of vaccine hesitancy among people [21]. According to the 3C model, researchers have roughly summarized the reasons for vaccine hesitancy in three dimensions: confidence, complacency, and constraints [22]. With the diversification and complexity of health information, Charles et al. added calculation and collective responsibility to the 3C model and proposed the 5C model [23]. Confidence (confidence in vaccines, healthcare workers, policymakers), complacency (not perceiving diseases as high risk), constraints (structural and psychological barriers), the calculation (engagements in extensive information searching), and collective responsibility (willingness to protect others) [24]. Tostrud et al. [25] argue that the 5C model is comprehensive in considering vaccine hesitancy causes. It can better match the complex information environment of social media and has also achieved good results in examining COVID-19 vaccine hesitancy [26]. Therefore, this study will use the 5C model to analyze the phenomenon and causes of Weibo users’ COVID-19 vaccine hesitancy [27].

In order to dig deeper into the causes of COVID-19 vaccine hesitancy in the Chinese population, this study will use the Latent Dirichlet Allocation (LDA, A computational model for automatic topic classification of text) model [28] to refine the themes and topics of the sample Weibos and conduct an artificial content analysis through the theoretical lens of the 5C model. Specifically, the research will be conducted around the following three research questions.

**RQ1.** How do Chinese people express vaccine hesitancy on Weibo? What themes and topics are involved in the statements?

**RQ2.** From the 5C model, what are the leading causes of vaccine hesitancy in China? What are the most severe causes?

**RQ3.** From the 5C model, have the causes of vaccine hesitancy among the Chinese changed over time? How have they changed?

## 2. Materials and Methods

### 2.1. Data Collection

COVID-19 was first discovered at the end of December 2019, and it has been spreading in China for more than three years since then [3]. Based on Weibo netizens’ discussions about the vaccine, this paper collected Weibos containing the keywords “(COVID-19 AND vaccine) AND (reject OR hesitate OR resist OR fear OR dare OR oppose)” between January 2020 and December 2022 (*n* = 5861). Then, by manual data cleaning, ads, emojis, and meaningless content were removed [19]. Finally, 3746 valid samples were obtained.

### 2.2. Topic Model

Topic modeling is an efficient text mining technique and a modeling method to extract implicit topics from large-scale texts effectively [29]. Latent Dirichlet Allocation (LDA), proposed by D.M. Blei et al. in 2003, is a three-layer Bayesian model based on probabilistic graphs that can refine keywords of texts and automatically cluster and group keywords [28]. It is faster in operation and is often used for topic extraction of short texts such as Weibo and Twitter [13]. In this study, we use LDA to find the themes and topics of the samples, as follows.

Step 1. LDA model building and grouping output of keywords. We used JGibbLDA to build the LDA model and analyze the sample. In the process, the researcher adjusted the number of the output groupings several times and observed the keywords output by LDA. It was found that the keywords in different groups had good mutual exclusivity when outputting 35 groups of topics.

Step 2. Examine the Topics of Weibos published at different times. By post-discrete analysis (applying LDA models on the entire sample set to obtain the topics while ignoring time first and then using the temporal information of the text to examine the distribution of Weibos in discrete time), we calculate which topic each Weibo belongs to and how it changes over time.

### 2.3. Content Analysis

The content analysis of this study is based on the results of LDA output [30]. As the 35 topics are too scattered to be discussed conveniently, we need to cluster them further. At the same time, we need to summarize the main reasons for each Weibo in the expressed vaccine hesitancy based on the output keyword results and the definition of the 5C model. In this study, confidence indicates users’ confidence in the COVID-19 vaccine, government, medical organizations, and vaccine development organizations; complacency indicates users’ contempt for Covid risk; constraints indicate users’ vaccination behavior is limited by their own psychological or physical conditions; calculation indicates the user’s search for and knowledge of vaccine information and the judgment of vaccination necessity obtained through information evaluation; and collective responsibility indicates the user’s willingness to protect others or promote herd immunity through vaccination [25,26,27]. The specific operations are as follows.

Step 3. Establishment of Topics and Themes names. We summarized the names of different Topics based on keywords. Moreover, we further divided the 35 Topics into 10 Themes based on the relevance and mutual exclusivity between the contents.

Step 4. Coding of the causes of vaccine hesitancy. Before formal coding, two coders were trained, and a randomly selected 10% of the sample was pre-coded. After the test was passed, the two coders coded all sample Weibos independently, marking the leading causes of vaccine hesitancy. During the coding process, if a Weibo were encountered to contain two or more reasons at the same time, the researcher would select one of them as the main reason based on the expression of the content. If a Weibo with unknown causes were encountered, they were labeled as Other.

Step 5: Reliability test for coding the causes of vaccine hesitancy. Inter-coder reliability scores were calculated using Scott’s Pi Coefficient (π) [31]. The scores all exceed 75%, indicating high coding reliability. When different opinions appeared, the coders chose a suitable one after discussion.

## 3. Results

### 3.1. COVID-19 Vaccine Hesitancy Expressed by Chinese on Weibo (RQ1)

Regarding our first research question, we found that the Chinese population expressed vaccine hesitancy bluntly on Weibo, with more lively discussions involving information access, vaccination services, physical illness, etc. (Table 1).

Information access (18.59%) is the most common theme of vaccine hesitancy, and Chinese people’s access to news in overseas media (“International News”, 6.43%) and during interpersonal communication (“Interpersonal Communication”, 5.26%) directly affects their vaccine hesitancy. For example, one Weibo stated, “Albert, the CEO of Pfizer USA, who had received four doses of Pfizer’s COVID-19 vaccine, still contracted New Crown virus, which had made them doubt the effectiveness of the COVID-19 vaccine. doubts”. The Chinese public often gets information about the new vaccine through media platforms, such as anti-vaccine posts on social media like Weibo (“Social Media”, 3.34%), public videos of vaccination refusals on online media (“Online Media”, 1.34%)”, and traditional media (“Traditional Media”, 0.51%) on news reports about vaccine safety. These messages influenced their value judgment about vaccination with the COVID-19 vaccine. In particular, the experts’ opinions (“Expert Opinion”, 1.71%) about the COVID-19 vaccine published on media platforms may have a counter effect on people’s willingness to be vaccinated. For example, some people said, “Experts’ vague explanations only smear more and more to make people afraid”.

Weibos involving vaccination services (13.91%) had the second highest percentage. A topic under this theme described the impact of medical staff on the vaccination process (“Medical staff”, 6.25%). Other topics describe the adverse effects of the sites of vaccination (“Vaccination sites”, 3.98%) and the schedule of vaccination (“Vaccination schedule”, 3.68%). For example, most of the population reported that they could not leave their job tasks during work to receive the COVID-19 vaccine.

Weibos about physical illness theme (13.24%) were also in the majority, with pre-existing underlying diseases preventing some people from receiving the COVID-19 vaccine. Among them, people with allergies were concerned about the possible physical reactions to the vaccine, as severe allergies could lead to shock (“Allergic Diseases”, 6.59%). Common sudden acute illnesses such as gastroenteritis and fever (“Acute Illness”, 3.66%), and chronic illnesses in the elderly such as hypertension and diabetes (“Chronic Diseases”, 2.99%) are also important factors that affect vaccination with the COVID-19 vaccine. For example, a public member said, “My mother is old and has high blood pressure and blood sugar, so I am afraid to take her to get the vaccine”.

The number of Weibos involving vaccine policy themes (12.96%) ranked fourth. A topic described macro-level national policy (“National Policy”, 3.04%) and suggested that mandatory vaccination has caused resentment and resistance among some Chinese people. For example, one person reported, “I’m tired of the daily phone calls from the community staff urging me to get the COVID-19 shots.” Some topics described the specific process (“Vaccination Process”, 6.83%) and work arrangement (“Vaccination Work”, 1.92%) of the COVID-19 vaccination. In addition, a small number of people expressed concern about the type of vaccine (“Vaccine Characteristics”, 1.17%).

In the risk perception (11.16%) theme, Weibos showed that the public’s “Social Perception” (4.70%) and “Self-awareness” (1.58%) influenced their risk judgment of vaccination. For instance, some people took a chance and believed their probability of contracting COVID-19 was very small. In terms of perception of COVID-19 disease, Chinese people’s perceptions of “Disease Severity” (4.00%) and “Disease Susceptibility” (0.88%) of COVID-19 also influenced their attitude toward vaccination. For example, some people thought COVID-19 was not severe and similar to influenza, so it did not matter if they did not get vaccinated.

The sixth theme is about special groups (10.63%). “Elderly People” (4.78%) and “Young Children” (3.63%) are two groups that must be considered for the promotion of the COVID-19 vaccination. Because of their age, some elderly people and children lack good self-care skills and health awareness.” Pregnant women” (2.22%) are another group that needs attention. Some pregnant women oppose the COVID-19 vaccine because they think it will cause miscarriage or fetal malformation.

Vaccine Side Effects (8.62%) mainly include “General Side Effects” (5.39%), such as dizziness, weakness, etc. Moreover, “Serious Side Effects” (3.23%), such as organ damage and metabolic reactions, may be associated with COVID-19 vaccination. The related side effects, although less likely to occur, are causing the population to resist the vaccine.

The vaccine effectiveness theme (6.56%) focuses on the Chinese population’s interest in the COVID-19 vaccine’s positive effects. Related topics describe the effectiveness of the vaccine in “Disease Prevention” (2.46%), “Infection Prevention” (1.68%), “Death Prevention” (0.61%), and “Inpatient Prevention” (0.37%). For example, some people did not believe that the COVID-19 vaccination prevented individuals from contracting COVID-19 and therefore chose to decline the vaccination. Other topics described “Vaccine Technology” (0.80%) and “Clinical Trials” (0.64%) of COVID-19 vaccination. For example, a breastfeeding woman stated, “There are no clinical trials for Breastfeeding mothers, and I am afraid to gamble with my child’s life”.

Psychological fear (2.43%) includes fear of needles for injections (“Needle Fear”, 1.82%) and fear of medical personnel (“Medical Fear”, 0.61%). For example, some people fear going to hospitals and may faint when they see white coats and needles.

Finally, the responsible attitude theme (1.90%) relates to the topics “Herd Immunity” (1.20%) and “Family Responsibilities” (0.70%). The lack of awareness of herd immunity or the lack of collective responsibility of some people leads to a Negative willingness to receive the COVID-19 vaccine. For example, some people felt that “the COVID-19 vaccine is irrelevant. It does not matter if we get it or not”.

### 3.2. Reasons for COVID-19 Vaccine Hesitancy in Chinese (RQ2)

Regarding our second research question, Table 1 first shows the main reasons under different Topics. It is easy to find that constraints, confidence, and calculation account for many reasons. Under the same topic, multiple reasons tend to co-exist as well. For example, the vaccine information provided by expert opinion tends to be more objective and comprehensive, suggesting the advantages of vaccination to the public and informing the vaccine’s side effects. On the contrary, experts overemphasize the benefits of vaccines and do not mention the side effects, which may lead to suspicion of experts and bring confidence-type vaccine hesitancy.

Based on the content analysis results, we further counted the number of Weibos for the causes of each type of vaccine hesitancy (Table 2). Constraints (35.48%) was one of the leading causes of vaccine hesitancy, mainly involving vaccination services, physical illness, special group, and psychological fear. Some public members could only receive vaccines after work or school, which developed vaccine hesitancy. More public members developed vaccine hesitancy because of physical (e.g., underlying illness or age problems) or psychological (e.g., needle or blood sickness) reasons. The number of people who could not receive vaccines because of physical problems was relatively higher.

Confidence (17.94%) is second only to constraints, mainly related to the themes of vaccine policy, side effects, and effectiveness. On the one hand, China has implemented a health policy of universal free vaccination against COVID-19 in the past three years. Various departments have been actively mobilizing the population for COVID-19 vaccination. Although many people get vaccinated because of the intense propaganda campaign, which has achieved mass immunization in a relatively short period, it has also led to a rebellious attitude among some people. On the other hand, China mainland is mainly vaccinated with the domestically produced COVID-19-inactivated vaccine. The domestic vaccine was put into use urgently at the early marketing stage before the completion of phase III clinical trials, which also led to some users’ concerns about the efficacy and side effects of the vaccine.

Calculation (15.99%) was the third most popular vaccine, mainly related to the information access theme. Some international news (e.g., Pfizer’s COVID-19 vaccine has a higher infection prevention rate than the domestic COVID-19 vaccine) disparaged the Chinese domestic COVID-19 vaccine and induced the public to weigh the cost-effectiveness of the vaccine they received. Interpersonal communication can quickly induce the public to weigh the pros and cons [32]. Some users who receive further advice from friends, family, and teachers need help judging and choosing the best vaccination option. In addition, social media, online media, and traditional media provide users with information about vaccines with their focus. It is easy to be ambivalent when users see the advantages of vaccines on TV and the adverse effects of vaccination on Weibo simultaneously.

Complacency (9.63%) was moderate, mainly related to the theme of risk perception. Some users were confident in their health condition, believing that they would not get COVID-19 or be cured by themselves quickly. Therefore, they refuse to receive the COVID-19 vaccine. These users tend to have a weak risk perception and a very determined attitude of vaccine hesitancy, which is difficult to change by external forces.

Collective responsibility (1.63%) has the lowest percentage of users, mainly related to the responsible attitude theme. The first type of users is those who oppose and question “herd immunization”, believing that herd immunization will not eliminate the COVID-19 virus, so they refuse vaccination. The second group of users is similar to those under complacency, who believe that even if infected, they will not cause problems for others.

Finally, 19.33% of Weibos are rated as other. These Weibos tend to be short, mostly in phrases or short sentences, and the content of Weibos expresses the situation of vaccine hesitancy. However, they do not reveal the specific reasons (e.g., I do not want to get the new vaccine). Weibos, in this section, is not in the minority. Although we cannot gain insight into the reasons for some users’ vaccine hesitancy, we can see that Weibo carries the function of expressing negative emotions of the Chinese vaccine hesitancy population.

### 3.3. Chronological Changes in Reasons for Vaccine Hesitancy in Chinese (RQ3)

COVID-19 has been disseminated in China for over three years. Moreover, vaccine hesitancy has occurred continuously in the population during this time. To answer our third research question, the researchers plotted a line graph of sample size over time and a bar graph of the proportion of causes of vaccine hesitancy over time (Figure 1).

Looking at the line graph, from 2020 to 2022, Weibos averages 105 entries per month, and the number fluctuates widely across months overall (σ = 107.73). In 2020, the number of Weibos was low (M_2020_ = 24, σ_2020_ = 14.64), and vaccine hesitancy received less discussion. In 2021, Weibos had the highest number (M_2021_ = 220, σ_2021_ = 110.04), and the number fluctuated dramatically. In particular, in April 2021, the number of Weibos expressing vaccine hesitancy was as high as 503. In 2022, the change in the number of Weibos leveled off again (M_2022_ = 69, σ_2022_ = 35.95), and the mean value of the number of Weibos was between 2020 and 2021.

From the bar chart, from 2020 to 2022, the proportion of vaccine hesitancy caused by constraints is getting higher, and the proportion of vaccine hesitancy caused by others is getting lower. Confidence and calculation have a more similar proportion distribution. Most of the time, the proportion of complacency decreases slightly over time, with the most significant proportion in 2020. While not as high, the proportion of Collective Responsibility increased significantly in 2022.

## 4. Discussion

This paper examines the expression of vaccine hesitancy on Weibo in China from 2020 to 2022. It uses the 5C model of vaccine hesitancy as the analytical lens. Topic analysis based on the LDA model and manual-based content analysis examines how Chinese people express their vaccine hesitancy on social media, the reasons for vaccine hesitancy, and the changes in the reasons for vaccine hesitancy over time.

The Chinese public expresses much content on Weibo involving COVID-19 vaccine hesitancy, involving a wide range of themes and topics. The causes are not only related to the users’ perceptions and conditions but are also influenced by multiple factors such as information exposure, health awareness, risk perception, and social identity [33]. The complexity of the causes further leads to the intersection of the content of the users’ discussions. In terms of themes, information access, vaccination services, physical illness, vaccine policy, and risk perception are closely related to vaccine hesitancy among the Chinese population. Users of vaccine hesitancy caused by information access reported that their refusal to vaccinate was related to information provided by international news, interpersonal communication, social media, and other channels. Regarding specific topics, topics such as allergic diseases, international news, medical staff, general side effects, and interpersonal communication were prevalent in expressing vaccine hesitancy. Some studies suggest that about 200–250 million Chinese people suffer from allergic diseases [34]. Thus, whether vaccines can cause allergies or other side effects is a concern for many people. In addition, topics such as international news and interpersonal communication were mentioned several times, suggesting a potential connection between the dissemination of vaccine information and vaccination efforts.

Constraints, confidence, and calculation are the leading causes of vaccine hesitancy among the Chinese public. The reasons for Chinese vaccine hesitancy differ slightly from those in some developed countries. While investigating the feelings of US people about COVID-19 vaccines, Thelwall et al. found that vaccine hesitators surveyed in that country blamed such vaccines for being rushed (37%) or simply mistrusting them (12%) [18,35]. In the UK, vaccine hesitancy is somewhat explained by mistrust toward the country’s healthcare system [36]. The Chinese government has implemented several initiatives to promote the COVID-19 vaccine, such as free vaccination, setting up vaccination points in schools and residential areas, and providing vaccination to the elderly at home [37]. However, some public members still need help to perceive the convenience of vaccination conditions due to work or study. In addition, many people with underlying diseases, pregnant women, babies, and older adults also have to develop vaccine hesitancy for medical reasons. Those who develop vaccine hesitancy because of their confidence are often skeptical about the safety and efficacy of vaccines and even rise to the level of skepticism of healthcare professionals, public health experts, governments, etc. Some users say, “The media and experts do not talk much about the side effects of vaccines, but many people around me have experienced dizziness and diarrhea after receiving vaccines.” Vaccine hesitancy resulting from the calculation is often difficult to change because it is a decision to be made by the vaccinator after careful consideration [38].

The causes of vaccine hesitancy in the Chinese population have changed dramatically over time. From 2020 to 2022, the proportion of constraints and collective responsibility increased; the proportion of complacency and others decreased; and the proportion of confidence and calculation changed relatively steadily. In 2020, many people could not made aware of the infectiousness and persistence of the COVID-19 virus, and vaccine hesitancy due to complacency was very common [39]. Coupled with the long development time of the COVID-19 vaccine, there was no available COVID-19 vaccine at the beginning of the outbreak, and some populations were influenced by multiple factors such as environment, body, and media, producing much vaccine hesitancy outside of 5C. At the end of December 2020, the Chinese domestic COVID-19 vaccine was officially launched, and there was also a large influx of vaccine hesitancy on Weibo [19]. Constraints, confidence, and calculation all have high proportions in 2021. Feedback from the sample shows that many public members have a wait-and-see attitude toward COVID-19 vaccination and need help to quickly develop a sense of trust in new vaccines [37]. People with underlying diseases are anxious and say they “cannot afford the unknown risks” of the COVID-19 vaccine to their bodies. During this period, there was a constant debate about the advantages and adverse effects of the vaccine. Some users were gradually swept up by negative information during the information access process, resulting in vaccine hesitancy caused by calculation.

Previous studies have also explored the phenomenon and causes of vaccine hesitancy in the Chinese population, but primarily cross-sectional surveys [40,41]. Based on them, this study focuses on the COVID-19 vaccination, extends the investigation time, and adopts a mixed study approach. This study can bring at least three research implications. (a) At the time of the complete deregulation of COVID-19 in China, a comprehensive description of the changes in COVID-19 vaccine hesitancy in the Chinese population over three years can help to combine the changes in the causes of vaccine hesitancy with the temporal changes to grasp the deeper situation of vaccine hesitancy. (b) Using a hybrid research method that combines machine-based topic modeling and human-based content analysis, we can efficiently and deeply explore the expression content and leading causes of vaccine hesitancy. The innovation of the method can provide technical references for other similar studies. (c) During the analysis, the researchers combined Weibos’ themes, topics, time, and 5C models to help deeply grasp the leading causes of vaccine hesitancy in the Chinese population. The findings can inspire public health professionals, medical practitioners, the government, and scientists how to intervene appropriately and reduce vaccine hesitancy.

There are also some shortcomings in this study. On the one hand, we collected the sample from Weibo, one of China’s largest social media, and examined the content of vaccine hesitancy. We can speculate several critical reasons for Chinese people’s vaccine hesitancy. However, we cannot rule out that some causes may be more obscure and not appear on social media; thus, we cannot know them. On the other hand, we only distilled the themes, topics, and reasons for vaccine hesitancy from our sample. In future studies, more findings may be obtained if the posters’ identity, education, and age are included in the examination.

## 5. Conclusions

China is a large country with a large population, and the causes of the population’s vaccine hesitancy are complex. This study examined the content of vaccine hesitancy, the causes of vaccine hesitancy, and the changes of causes over time expressed by Chinese people on Weibo from 2020 to 2022. The results show that when the Chinese expressed vaccine hesitancy, it usually involved themes such as information access, vaccination services, and physical illness, and topics such as the vaccination process, allergic diseases, and international news. Constraints, confidence, and calculation are the leading causes of vaccine hesitancy on Weibo. Therefore, the government and public health organizations need to provide the public with convenient vaccination conditions as much as possible, enhance public trust in vaccines, and reduce the cost of vaccination for the people in their efforts to promote vaccines. The study’s results can help inspire public health experts, health organizations, or governments in various countries to improve the vaccine hesitancy phenomenon and enrich the practical experience of vaccination worldwide.

## Figures and Tables

**Figure 1 vaccines-11-00712-f001:**
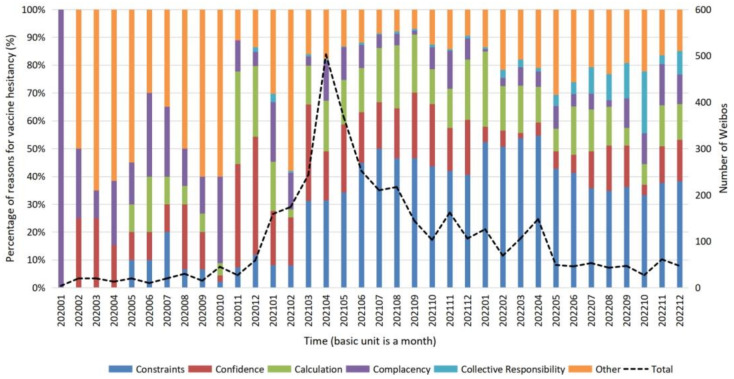
Changes in the number of Weibos and reasons for vaccine hesitancy over time.

**Table 1 vaccines-11-00712-t001:** Themes and Topics in Sample Weibos.

Theme and Topic Labels	Number (%)	Top Terms (Stemmed)	Main Reasons for Vaccine Hesitancy (5C)
**Theme: Information Access**	**696 (18.59%)**		
“International News”	241 (6.43%)	Pfizer, effects, US, UK, infection	Calculation/Other
“Interpersonal Communication”	197 (5.26%)	Friends, teachers, students, informants, family	Calculation/Other
“Social Media”	125 (3.34%)	Weibo, Tiktok, video, post, retweet	Calculation
“Expert Opinion”	64 (1.71%)	Doctors, specialists, authorities, respiratory, internal	Calculation/Confidence
“Online Media”	50 (1.34%)	Baidu, search, find, experience, online	Calculation
“Traditional Media”	19 (0.51%)	TV, CCTV, coverage, official, advocacy	Calculation
**Theme: Vaccination Services**	**521 (13.91%)**		
“Medical Staff”	234 (6.25%)	Doctors, nurses, injections, time, hard work	Confidence/Constraints
“Vaccination Sites”	149 (3.98%)	Hospital, venue, temporary, taxi, build	Constraints
“Vaccination Schedule”	138 (3.68%)	Work, school, time-consuming, special, evenings	Constraints
**Theme: Physical Illness**	**496 (13.24%)**		
“Allergic Diseases”	247 (6.59%)	Allergy, edema, neurological, shock, reaction	Constraints
“Acute Illness”	137 (3.66%)	Heart attack, brain attack, bowel, failure, fever	Constraints
“Chronic Diseases”	112 (2.99%)	Hypertension, diabetes, cancer, kidney, uremia	Constraints
**Theme: Vaccine Policy**	**486 (12.96%)**		
“Vaccination Process”	256 (6.83%)	Three stitches, months, strengthen, optimise, enhance	Confidence/Other
“National Policy”	114 (3.04%)	Provide, free, universal, mandatory, health insurance	Confidence/Other
“Vaccination Work”	72 (1.92%)	Community, organization, compulsory, unit, disinfection	Confidence/Other
“Vaccine Characteristics”	44 (1.17%)	Inactivation, storage, convenience, technology, backwardness	Confidence/Other
**Theme: Risk Perception**	**418 (11.16%)**		
“Social Perception”	176 (4.70%)	Mask, droplets, socialization, distance, not necessarily	Complacency/Constraints
“Disease Severity”	150 (4.00%)	Influenza, asymptomatic, foreign, liberalized, mild	Complacency
“Self-awareness”	59 (1.58%)	Body, strong, no worries, quick, recovery	Complacency
“Disease Susceptibility”	33 (0.88%)	Isolation, China, new, security, hotels	Complacency
**Theme: Special Groups**	**398 (10.63%)**		
“Elderly People”	179 (4.78%)	Age, old, vulnerable, health, foreknowledge	Constraints/Confidence
“Young Children”	136 (3.63%)	Young, infant, unable, rest, vascular	Constraints/Confidence
“Pregnant Women”	83 (2.22%)	Pregnancy, miscarriage, bleeding, danger, fear	Constraints/Confidence
**Theme: Vaccine Side Effects**	**323 (8.62%)**		
“General Side Effects”	202 (5.39%)	Dizziness, weakness, soreness, worry, after-effects	Confidence
“Serious Side Effects”	121 (3.23%)	Organ, allergic reaction, pruritus, abnormality, damage	Confidence
**Theme: Vaccine Effectiveness**	**246 (6.56%)**		
“Disease Prevention”	92 (2.46%)	Safe, no, severe, most, mild	Confidence
“Infection Prevention”	63 (1.68%)	Infection, breakthrough, prevention, effect, nucleic acid	Confidence
“Vaccine Technology”	30 (0.80%)	First generation, traditional, outdated, research, competence	Confidence/Other
“Clinical Trials”	24 (0.64%)	Invalid, test, no, paper, true	Confidence/Other
“Death Prevention”	23 (0.61%)	Valid, certificate, few, deaths, cases	Confidence
“Inpatient Prevention”	14 (0.37%)	Emergency, hospitalization, causes, statistics, decline	Confidence
**Theme: Psychological Fear**	**91 (2.43%)**		
“Needle Fear”	68 (1.82%)	Needle, sharp, stinging, dizzy, frightened	Constraints/Confidence
“Medical Fear”	23 (0.61%)	White coat, blood pressure, fainting, arm, smell	Constraints
**Theme: Responsible Attitude**	**71 (1.90%)**		
“Herd Immunity”	45 (1.20%)	No, relationship, immunity, barrier, scam	Collective Responsibility
“Family Responsibilities”	26 (0.70%)	Single, indifferent, children, voluntary, testing	Collective Responsibility/Confidence
**Total**	**3746 (100.00%)**		

**Table 2 vaccines-11-00712-t002:** Reasons for COVID-19 Vaccine Hesitancy.

Reasons for Vaccine Hesitancy (5C)	Number (%)	Cumulative Number (%)
Constraints	1329 (35.48%)	1329 (35.48%)
Confidence	672 (17.94%)	2001 (53.42%)
Calculation	599 (15.99%)	2600 (69.41%)
Complacency	361 (9.63%)	2961 (79.04%)
Collective Responsibility	61 (1.63%)	3022 (80.67%)
Other	724 (19.33%)	3746 (100%)
Total	3746 (100%)	3746 (100%)

## Data Availability

The datasets generated and/or analyzed during the current study are not publicly available due to the potential sensitivity of the content, but they are available from the corresponding author upon reasonable request.

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
