# Peer review of "COVID-19 Vaccine Hesitancy in China: An Analysis of Reasons through Mixed Methods"

_vaccines, 2023, doi:10.3390/vaccines11030712_

Round 1

Reviewer 1 Report

The LDA model and content analysis were used to analyze the content of COVID-19 vaccine hesitancy expressed by the Chinese on Weibo from 2020 to 2022, the leading causes of vaccine hesitancy, and the changes in the reasons for vaccine hesitancy over time. LDA abbreviation should be explained

In the abstract at least briefly should be reported quantitative results of thye items investigated 

Due to the continued global spread of COVID-19 and the lack of an ideal drug, vaccination has been seen as the primary means of preventing and controlling COVID-19 in many countries [1] ...add with high safety profile and include other references, as follows

10.1007/s10900-022-01082-8

 In order to dig deeper into the causes of COVID-19 vaccine hesitancy in the Chinese population, this study will use the Latent Dirichlet Allocation (LDA) model to refine the themes and topics of the sample Weibos and conduct an artificial content analysis through the theoretical lens of the 5C model. Specifically, the research will be conducted around the following three research questions. RQ1. How do Chinese people express vaccine hesitancy on Weibo? What themes and 67 topics are involved in the statements? RQ2. From the 5C model, what are the leading causes of vaccine hesitancy in China? What are the most severe causes? RQ3. From the 5C model, have the causes of vaccine hesitancy among the Chinese changed over time? How have they changed? thios part will be fit better in the methods section instead of the introduction

discussion can be shortened

impact of the study should be underlined in the conclusions, as well as with indications of specific strategies to address vaccien hesitancy based on the obtained results

Author Response

Comment 1. The LDA model and content analysis were used to analyze the content of COVID-19 vaccine hesitancy expressed by the Chinese on Weibo from 2020 to 2022, the leading causes of vaccine hesitancy, and the changes in the reasons for vaccine hesitancy over time. LDA abbreviation should be explained.
Response 1. Thank you very much for your review of this article. With your suggestion, We have added an explanation of the LDA.

Comment 2. In the abstract at least briefly should be reported quantitative results of thye items investigated.
Response 2. Based on your suggestion, we have added some critical quantitative results in the abstract.

Comment 3. Due to the continued global spread of COVID-19 and the lack of an ideal drug, vaccination has been seen as the primary means of preventing and controlling COVID-19 in many countries [1] ...add with high safety profile and include other references, as follows 10.1007/s10900-022-01082-8.
Response 3. Based on your suggestions, we have revised some references.

Comment 4. “In order to dig deeper into the causes of COVID-19 vaccine hesitancy in the Chinese population, this study will use the Latent Dirichlet Allocation (LDA) model to refine the themes and topics of the sample Weibos and conduct an artificial content analysis through the theoretical lens of the 5C model. Specifically, the research will be conducted around the following three research questions. RQ1. How do Chinese people express vaccine hesitancy on Weibo? What themes and 67 topics are involved in the statements? RQ2. From the 5C model, what are the leading causes of vaccine hesitancy in China? What are the most severe causes? RQ3. From the 5C model, have the causes of vaccine hesitancy among the Chinese changed over time? How have they changed?” this part will be fit better in the methods section instead of the introduction.
Response 4. After our extensive discussions and examination of the published articles on vaccines, we decided to place the research question at the end of the introduction.

Comment 5. Discussion can be shortened.
Response 5. Based on your suggestion, we removed some unnecessary content (mainly some cases or descriptive content) in the discussion section. And We added parts of more meaningful content (e.g., when discussing the reasons for vaccine hesitation, we compared China and other countries. And reflections on the lack of research have been added).

Comment 6. Impact of the study should be underlined in the conclusions, as well as with indications of specific strategies to address vaccien hesitancy based on the obtained results.
Response 6. Based on your suggestions, we have added some strategies to address vaccine hesitancy in the conclusion section. It has made our conclusions more insightful. Thank you very much for your valuable suggestions.

Reviewer 2 Report

I  do not have a comments. In my opinion this is a very interesting paper.

Author Response

Comment. I do not have a comments. In my opinion this is a very interesting paper.
Response. Thank you very much for reviewing and receiving this article.

Reviewer 3 Report

I was invited to revise the paper entitled "COVID-19 Vaccine Hesitancy in China: An Analysis of Reasons through Mixed Methods". It aimed to evaluated the vaccine hesitancy among users of the most used social network among chinese (Weibo) via 5c models.

The methodology is innovative and appropriate for this study and results are clear and deeply described.

My major concer is about the selection bias. Non-hesitant people don't use social media to get informed about vaccines. In my opinion, these results are larger than they are in reality.

In addition, this kind of analysis doesn't take into accoutn the level of instruction and the socioeconomical status. This kind of discussion on social media frequently were performed by people with low level of instruction.

Among discussion, no comparisons with vaccine hesitancy from other country was performed.

In addition, strenght and limitation section was not reported.

Author Response

Comment 1. I was invited to revise the paper entitled "COVID-19 Vaccine Hesitancy in China: An Analysis of Reasons through Mixed Methods". It aimed to evaluated the vaccine hesitancy among users of the most used social network among chinese (Weibo) via 5c models. The methodology is innovative and appropriate for this study and results are clear and deeply described.
Response 1. Thank you very much for your review of this article. With your suggestions, we have made sufficient changes to the article's content and revised some language issues.

Comment 2. My major concer is about the selection bias. Non-hesitant people don't use social media to get informed about vaccines. In my opinion, these results are larger than they are in reality. In addition, this kind of analysis doesn't take into accoutn the level of instruction and the socioeconomical status. This kind of discussion on social media frequently were performed by people with low level of instruction.
Response 2. Following your suggestion, we have added statistics on Weibo users' level of instruction and evidence that social media can be used to express vaccine hesitancy in the Introduction. In the last paragraph of the Discussion, we also reflect on possible research shortcomings introduced by selecting the samples (or platform).

Comment 3. Among discussion, no comparisons with vaccine hesitancy from other country was performed. In addition, strenght and limitation section was not reported.
Response 3. We compared the reasons for vaccine hesitancy in China, the United Kingdom, and the United States in the third paragraph of the Discussion. The article's strengths are reported in the second-to-last paragraph of the Discussion. The article's shortcomings are reported in the bottom paragraph of the Discussion. These revisions make our Discussion more profound. Thank you very much for your valuable suggestions.

Round 2

Reviewer 3 Report

It can be accepted for publication